# Resistome diversity in cattle and the environment decreases during beef production

**Noelle R Noyes[1]\*, Xiang Yang[2], Lyndsey M Linke[1], Roberta J Magnuson[1], Adam Dettenwanger[3], Shaun Cook[4], Ifigenia Geornaras[2], Dale E Woerner[2], Sheryl P Gow[5], Tim A McAllister[4], Hua Yang[2], Jaime Ruiz[3], Kenneth L Jones[6], Christina A Boucher[3], Paul S Morley[1]†, Keith E Belk[2]†**

[1]Department of Clinical Sciences, Colorado State University, Fort Collins, United States; [2]Department of Animal Sciences, Colorado State University, Fort Collins, United States; [3]Department of Computer Sciences, Colorado State University, Fort Collins, United States; [4]Agriculture and Agri-Food Canada Research Centre, Lethbridge, Canada; [5]Centre for Food-borne, Environmental Zoonotic Infectious Diseases, Public Health Agency of Canada, University of Saskatoon, Saskatoon, Canada; [6]Department of Biochemistry and Molecular Genetics, University of Colorado Denver School of Medicine, Aurora, United States

**\*For correspondence:** noelle.
noyes@colostate.edu

†These authors contributed
equally to this work

**Competing interests:** The
authors declare that no
competing interests exist.

**Reviewing editor:** Ben Cooper,
Mahidol Oxford Tropical
Medicine Research Unit, Thailand

**Abstract** Antimicrobial resistant determinants (ARDs) can be transmitted from livestock systems through meat products or environmental effluents. The public health risk posed by these two routes is not well understood, particularly in non-pathogenic bacteria. We collected pooled samples from 8 groups of 1741 commercial cattle as they moved through the process of beef production from feedlot entry through slaughter. We recorded antimicrobial drug exposures and interrogated the resistome at points in production when management procedures could potentially influence ARD abundance and/or transmission. Over 300 unique ARDs were identified. Resistome diversity decreased while cattle were in the feedlot, indicating selective pressure. ARDs were not identified in beef products, suggesting that slaughter interventions may reduce the risk of transmission of ARDs to beef consumers. This report highlights the utility and limitations of metagenomics for assessing public health risks regarding antimicrobial resistance, and demonstrates that environmental pathways may represent a greater risk than the food supply.

## Introduction

Food production and food products are important potential sources of antimicrobial resistant (AMR) infections in humans. Beef is a widely consumed protein commodity, and production and consumption is expected to increase in the United States and globally (*Daniel et al., 2011*; *OECD/Food and Agriculture Organization of the United Nations, 2015*). In North American beef production, several critically important antimicrobial drugs (AMDs) such as fluoroquinolones, macrolides and third-generation cephalosporins are used, while others are not, e.g., carbapenems (*World Health Organization, 2011*; *Food and Drug Administration, 2003*). Use of these AMDs is thought to increase the risk of AMR being transmitted to humans through environmental exposures (i.e., air, water and soil), occupational exposures (*Levy et al., 1976*; *Moon et al., 2015*), as well as through consumption of beef products (*Antibiotic Resistance from the Farm to the Table [Internet], 2014*). While surveillance for foodborne AMR pathogens has been part of North American food safety systems for decades (*National Antimicrobial Resistance Monitoring System – Enteric Bacteria (NARMS), 2011*;

**eLife digest** When bacteria become resistant to antibiotics, it becomes difficult or impossible to treat infections in both people and animals. Antibiotic resistance is a growing problem, and many fear a "post-antibiotic era" in which common infections become life threatening.

In order to slow the spread of antibiotic resistance, it is important to understand how and where this resistance develops. In general, using antibiotics increases the likelihood that bacteria will develop resistance. Therefore, locations where antibiotics are commonly used – such as hospitals, long-term care facilities, livestock facilities (such as feedlots) and crop production areas (such as orchards) – may help antibiotic resistance to develop and spread. However, it is largely unknown how much each location promotes the emergence of antibiotic-resistant bacteria.

Until recently, we have only been able to investigate how resistance develops in bacteria grown in a laboratory; or to look for a handful of specific resistance genes in a sample of bacteria collected from people, animals or the environment. Fortunately, a technology called next-generation sequencing now allows us to look at all the resistance genes within all the bacteria in a sample. This may help us to improve our understanding of how and where resistance develops and spreads.

Noyes et al. have now used next-generation sequencing to describe the antibiotic resistance potential (known as the "resistome") found in various types of samples collected from feedlots and slaughterhouses involved in producing beef. This showed that the number of different resistance genes in the samples decreased while cattle were in the feedlot and during the slaughter process. Several groups of resistance genes that were detected when the cattle first arrived in the feedlot were not detected at all at the end of the feedlot period. However, some resistance genes were detected throughout the feedlot period, and these tended to be resistance genes that allow the bacteria to evade the same antibiotics that were used in the cattle. In addition, no resistance genes of any type were detected in the samples collected after the cattle had been slaughtered.

As well as providing insights into the resistome of beef production, Noyes et al.'s study also highlights the fact that we need to develop a deeper understanding of the data that come from next-generation sequencing. This may involve developing new laboratory techniques and creating new methods to analyze such data.

*Canadian Integrated Program for Antimicrobial Resistance Surveillance (CIPARS), 2013*), we have yet to fully understand and quantify the public health risk posed by transmission of non-pathogenic bacteria that carry antimicrobial resistance determinants (ARDs). These ARDs could pose a risk to human health if the bacteria carrying them become established within the microbiome of the human host, subsequently enabling horizontal gene transfer to pathogens (*Forsberg et al., 2012*; *Rolain, 2013*); or if these ARDs are present in opportunistic pathogens that become established within an immunocompromised individual. Establishment within an individual's microbiome could occur either through the ingestion of contaminated food products or through exposure to environmental effluents disseminated from beef feedlots (*Antibiotic Resistance from the Farm to the Table [Internet], 2014*), i.e., a facility where cattle are aggregated, reared in pens (i.e., outdoor enclosures) and fed a high-energy ration before being slaughtered. The rate at which ARDs from beef products or production facilities become established within humans is unknown, largely due to an historical reliance on culture and isolation of pathogens and an inability to access the microbial community and its complete repertoire of ARDs (i.e., the resistome).

Several steps in the beef production system could play crucial roles in the transmission of AMR from beef production to humans via environmental interfaces and beef products. In North America, use of AMDs is much greater in feedlots than any other phase of beef production (*Gow et al., 2008*; *Rao et al., 2010*), a fact that has raised concerns that these operations could represent the principal phase of beef production in which AMR is acquired or maintained. Furthermore, feedlots are intricately linked to environmental exposure pathways such as air, manure, soil, and water, enabling indirect human exposure to feedlot effluents (*McEachran et al., 2015*). In North America, abattoirs (i.e., slaughterhouses) are a potential control point for the transmission of AMR , as they employ sequential antibacterial interventions to reduce pathogen contamination in beef products; testing of these

interventions has demonstrated that they are effective in reducing not only pathogens, but also total bacterial contamination of beef products (*Bacon et al., 2000*). We hypothesized that the antimicrobial interventions and other procedures used in feedlots and abattoirs would exert a measurable effect on the presence, abundance and composition of ARDs in the bacterial populations of cattle, the feedlot environment and market-ready beef products. Furthermore, we hypothesized that the use of a metagenomics approach would enable us to quantify these changes at an ecological level and therefore better understand the risk to public health, compared to the use of a culture-based approach.

In order to understand how feedlots and abattoirs affect the transmission of ARDs , it is imperative to track pens of cattle through the beef production system (*Figure 1*), documenting AMD use and antimicrobial interventions and describing resistome changes over time. However, research in this area has been constrained by the challenges of tracking beef products and environmental effluents from individuals or pens of cattle, and collecting detailed records of AMD exposures for the cattle being studied. Specific challenges include lack of unique animal identification, use of non-computerized or hard-to-access AMD treatment records, effluents that are difficult to trace (e.g., air and runoff water), disassembly of carcasses into hundreds of non-linked parts, and the sheer difficulty of obtaining relevant, representative samples from feedlot steers, which weigh over 450 kg. Because of these complexities, studies in this area have been constrained to descriptions of AMR prevalence in isolated sectors of the beef production process without access to relevant AMD exposure data (*Sheikh et al., 2012*); or they have relied on the use of AMD and AMR data at very abstract levels such as the nation-state (*Chantziaras et al., 2013*). To our knowledge, no studies have specifically tracked antimicrobial use in cattle while investigating antimicrobial resistance in market-ready products or consumers. This dearth of evidence greatly complicates efforts to develop effective policies related to antimicrobial use in livestock with the goal of protecting public health (*Landers et al., 2012*). The objective of this study was to perform a prospective longitudinal analysis of antimicrobial use and resistance in beef production and to exploit shotgun metagenomics to characterize resistome dynamics in the environment and the products of cohort cattle from feedlot through to market-ready product.

## Results and discussion

A convenience sample of beef feedlots in Texas (n=2) and Colorado (n=2) was selected for study enrollment based on their willingness to participate. Capacity in these feedlots was 69,000, 73,000, 74,000, and 98,000 cattle, which is typical for large U.S. feedlots. While small feedlots (defined as feeding <100 cattle) comprise >60% of the ~26,500 feedlot operations in the U.S., large feedlots (defined as feeding >1000 cattle) contain >75% of all cattle that are reared in feedlots in the U.S. (*Census of Agriculture, 2012*). Animal handling and management procedures in study feedlots were typical for large feedlot operations in North America, and included daily observations of all pens by feedlot personnel trained to identify clinically ill cattle. Sick cattle were moved from their home pen to the cattle handling chute and/or a hospital pen to receive treatment. Rations in these 4 feedlots were corn-based and conformed to the National Research Council diet requirements for beef cattle (*National Research Council, 2000*). In each feedlot, 2 pens of cattle were enrolled in the study. Because cattle are typically raised in feedlots for at least 5 months, pens of newly arrived cattle are not always available for study participation. Therefore, study pens were a convenience selection from all pens that had been most recently filled to capacity at or near the start of the study. Feedlot owners and managers were aware of the identity of enrolled pens, but personnel involved in managing the cattle on a daily basis were not and therefore it is unlikely that standard operating procedures were altered during the study period. Pen capacities ranged from 150 to 281 cattle, and a total of 1741 cattle were housed in these 8 pens. Antimicrobial exposures were recorded throughout the feeding period, i.e., the total time that animals were housed in the feedlot, which ranged from 117 to 227 days (see *Figure 1—source data 1*). This duration is typical for North American beef cattle that enter feedlots at this age/weight. Analysis of these records indicates that all cattle received in-feed macrolides (tylosin), but use of parenteral AMDs was relatively infrequent at the individual animal level (*Table 1*). All pens contained at least one animal that was exposed to individual doses of parenteral tetracyclines and macrolides.

In order to assess the resistome throughout the feedlot and slaughter processes, we collected pooled, ecological-level samples from pens of cattle and their environment as they moved through the beef production system (*Figure 1*). Pooled samples provide a representative resistome for the unit of animal management in feedlots and abattoirs, i.e., the pen groupings; they do not provide insight into individual animal resistome variability, and therefore inferences in this study were made at the group level. Pooled fecal, soil and drinking water samples from each pen were collected within 6 weeks after the arrival of cattle at the feedlots ('arrival samples', n=24, one pooled sample per pen per sample type; see *Figure 1—source data 1*), as well as within 1 day before the same pens of cattle were shipped for slaughter ('exit samples', n=24, one pooled sample per pen per sample type). Cattle were then transported (<8 hr transport time) by truck to 2 abattoirs (one in Texas and one in Colorado) and the walls, floors and ceilings of the trucks were swabbed ('truck samples', n=8, one per pen) immediately after the cattle were unloaded; each abattoir slaughtered ~5500 cattle per day. Outside of the abattoirs, cattle were placed in holding pens (i.e., outdoor enclosures that can hold hundreds of cattle for a short amount of time), where pooled fecal and drinking water samples were collected after cattle had been moved into the abattoir ('holding samples', n=16, one per pen per sample type). Cattle were then euthanized and the carcasses disassembled into beef products. Both abattoirs employed multiple-hurdle interventions to reduce pathogen and other bacterial contamination on carcasses; this included hot water pasteurization, lactic and peroxyacetic acid spray, as well as knife trimming and spot steam vacuuming of the carcass (*Figure 1*). At the end of this process, pooled swab samples were taken from the conveyor belt used to transport

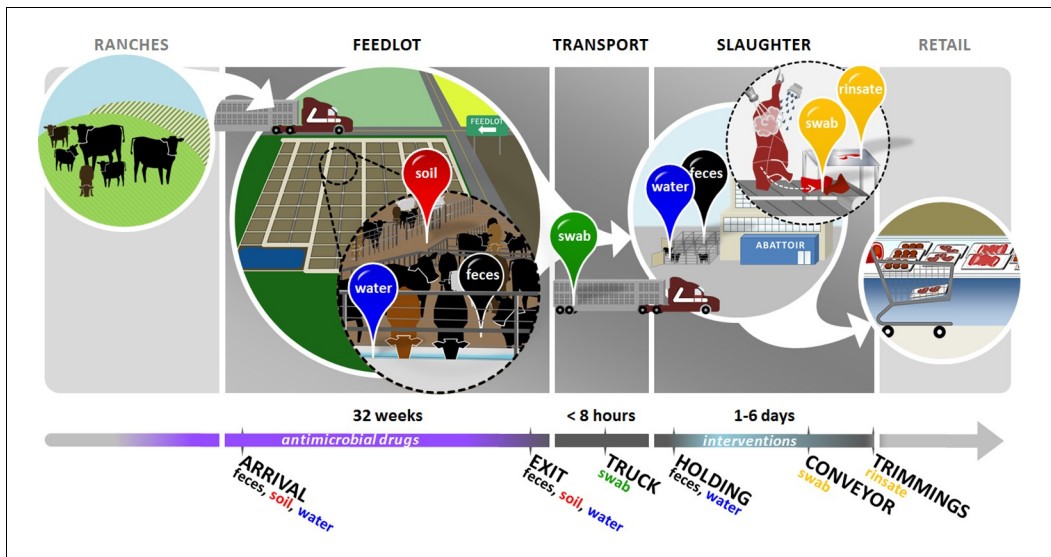

**Figure 1.** Overview of sampling design. Cattle in this study were born on ranches and entered the feedlots between 3 and 12 months of age. In the feedlots, we collected pooled fecal (black pin), soil (red pin), and drinking water (blue pin) samples from 2 pens of cattle in each of 4 feedlots. These samples were collected once around the time that study cattle arrived in the feedlot ('arrival'), and then once when the same cattle had reached slaughter weight and were ready to exit the feedlot ('exit'). Study cattle were then loaded onto transport trucks for shipment to the abattoir. Pooled swabs (green pin) from the inside walls of the transport trucks were collected immediately after the cattle had been unloaded at the abattoir ('truck'). Cattle were then placed into a holding pen outside of the abattoir, where pooled fecal (black pin) and drinking water (blue pin) samples were collected ('holding'). Cattle then entered the abattoir, where they were humanely slaughtered and their carcasses disassembled into beef products for retail. At the end of this process, we collected swabs (yellow pin) from the conveyor belts used to move carcass parts ('conveyor'), as well as rinsates (yellow pin) of the carcass trimmings used to make ground beef ('trimmings'). See *Figure 1—source data 1* for sampling details, including exact sampling dates for all 8 pens in this study.

The following source data is available for figure 1:

**Source data 1.** Sample collection details, by location, sample matrix and pen.

**Table 1.** Antimicrobial drug usage in the study population.

| Drug (dosage) | Drug Class | Primary Reason for Use | Number of Animals treated (%) | | | | | | | |
|---|---|---|---|---|---|---|---|---|---|---|
| | | | Pen A | Pen B | Pen C | Pen D | Pen E | Pen F | Pen G | Pen H |
| Tylosin phosphate (11 mg/kg diet dry matter)[a] | Macrolide | Liver abscess prevention | 244 (100) | 281 (100) | 152 (100) | 189 (100) | 230 (100) | 230 (100) | 265 (100) | 150 () |
| Tulathromycin[b] (2.5 mg/kg BW[c]) | Macrolide | BRD[d] Treatment | 15 (6.1) | 16 (5.7) | 12 (7.9) | 3 (1.6) | 19 (8.3) | 3 (1.3) | 8 (3.0) | 5 (3.3) |
| Oxytetracycline[e] (20 mg/kg BW) | Tetracycline | BRD Treatment | 1 (0.4) | 1 (0.4) | 43 (28.3) | 9 (4.8) | 6 (2.6) | 2 (0.9) | 13 (4.9) | 10 (6.7) |
| Oxytetracycline and Flunixin meglumine[b] (30 mg/kg BW and 2 mg/kg BW) | Tetracycline | BRD Treatment | 0 (0.0) | 0 (0.0) | 0 (0.0) | 0 (0.0) | 1 (0.4) | 0 (0.0) | 0 (0.0) | 0 (0.0) |
| Danofloxacin mesylate[b] (8 mg/kg BW) | Fluoroquinolone | BRD Treatment | 0 (0.0) | 4 (1.4) | 1 (0.7) | 0 (0.0) | 4 (1.7) | 7 (3.0) | 2 (0.8) | 0 (0.0) |
| Enrofloxacin[b] (7.7 mg/kg BW) | Fluoroquinolone | BRD Treatment | 0 (0.0) | 0 (0.0) | 0 (0.0) | 0 (0.0) | 0 (0.0) | 0 (0.0) | 2 (0.8) | 0 (0.0) |
| Ceftiofur sodium[e] (1 mg/kg BW) | β-lactam | BRD Treatment | 0 (0.0) | 0 (0.0) | 2 (1.3) | 0 (0.0) | 0 (0.0) | 0 (0.0) | 0 (0.0) | 0 (0.0) |
| Ceftiofur crystalline free acid[b] (6.6 mg/kg BW) | β-lactam | BRD Treatment | 0 (0.0) | 0 (0.0) | 0 (0.0) | 0 (0.0) | 0 (0.0) | 0 (0.0) | 4 (1.5) | 1 (0.7) |

[a]This AMD was in all rations of all cattle for the duration of the feeding period

[b]Each treated animal received a dose that persisted in target tissues at effective therapeutic concentrations for 3 days, according to the drug label.

[c]BW = body weight

[d]BRD = bovine respiratory disease

[e]Each treated animal received a dose that persisted in target tissues at effective therapeutic concentrations for 1 day, according to the drug label.

disassembled market-ready carcass parts (n=8, one per pen). In addition, beef trimmings (i.e., the parts of the carcass used to make ground beef) were collected and rinsed to obtain a pooled sample of the parts of the carcass with the highest food-safety risk (n=8, one per pen). The metagenomic sequences obtained from table and trimming samples represented the microbiome and resistome after antibacterial interventions had been applied to the carcass, and just before the beef products were packaged for retail distribution ('market-ready samples'). All sampling locations represented points in the beef production process when cattle or their end-products were actively managed and/or antimicrobial interventions were applied, both of which could influence the abundance and/or transmission of ARDs.

Total DNA was extracted from 88 samples and sequenced on an Illumina HiSeq, resulting in 407.7 Gb of sequence data (average 46.3 M reads per sample, range 12.0–93.4 M, *Supplementary file 1*). One drinking water sample did not contain enough DNA (i.e., <1 ng) to be sufficiently sequenced. Reads were trimmed and filtered for quality, and reads classified as host genome (*Bos taurus)* were removed from further analysis (*Supplementary file 1*). Non-host reads were then aligned to a custom non-redundant database of ARD sequences compiled from publicly available sources. ARDs with a gene fraction of >80% across all alignments were considered to be positively identified in a sample. We identified 319 unique ARDs across all 87 samples (*Supplementary file 2*), representing 1.2 M individual read alignments (*Figure 2—source data 1*). Most ARDs were present in low numbers (*Figure 2A*), and the proportion of trimmed, non-host reads aligning to ARDs in the database was correspondingly low across all samples (range 0.00% to 0.12%, *Supplementary file 1*). The median number of unique ARDs identified per sample was 33 (range: 0 to 136; *Figure 2B*). While this may be an underestimate due to sequencing coverage, recent studies using functional metagenomic screening of dairy cattle feces reported a maximum of 26 unique ARDs per animal (*Wichmann et al., 2014*). Functional metagenomics offers increased sensitivity compared to shotgun metagenomics (*Forsberg et al., 2014*), and therefore our observation of a relatively large number of unique ARDs per sample suggests that the pooling of samples undertaken in this study may have increased sensitivity. For comparison, a shotgun metagenomic study of 252 individual human fecal samples identified an average of 21 unique ARDs per sample (*Forslund et al., 2013*). The 319 ARDs identified in this study represented 42 AMR mechanisms

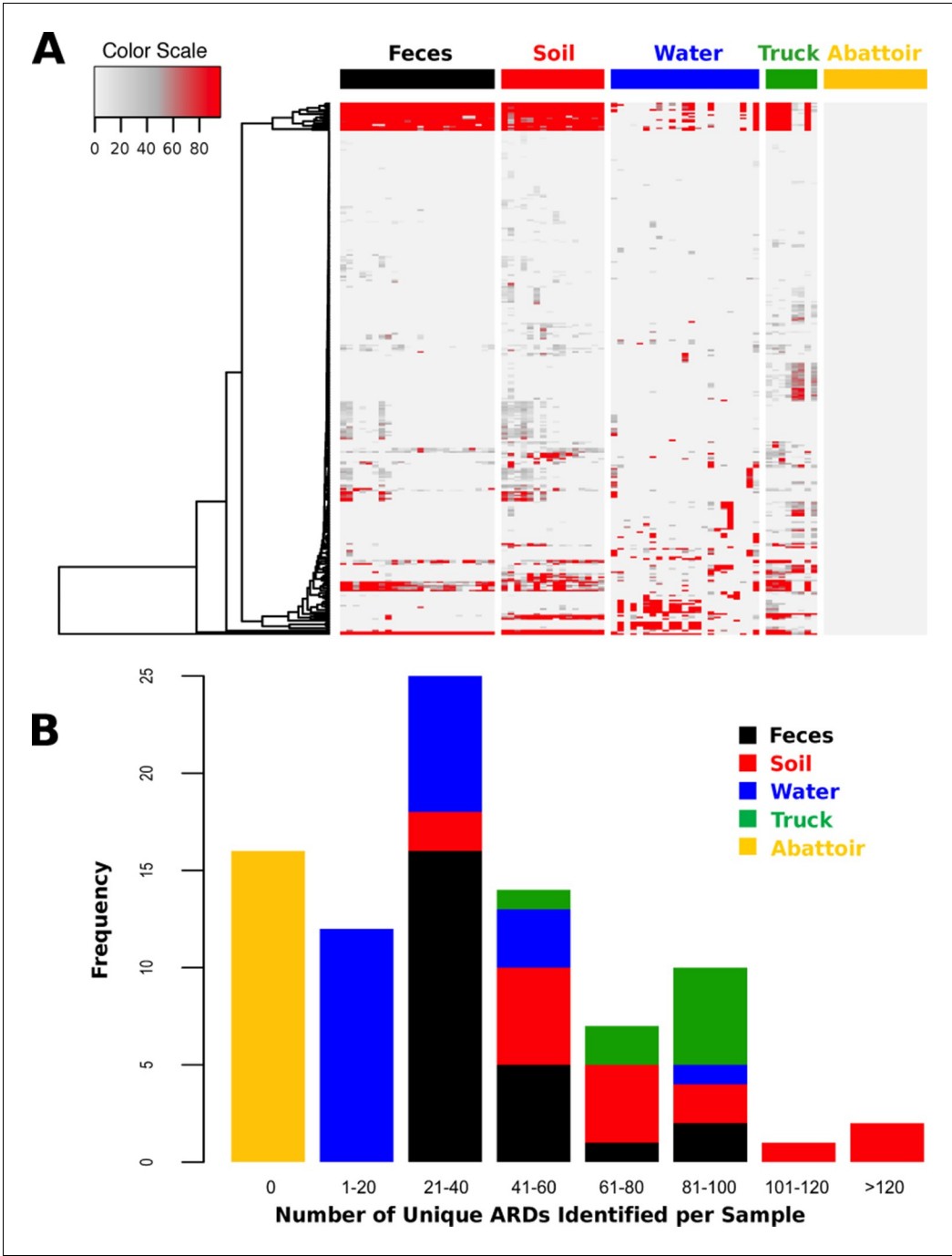

**Figure 2.** ARD abundance and frequency, by sample type. (**A**) Heatmap of the 319 ARDs (rows) identified in 87 samples (columns) collected from the beef production system. Columns are grouped by sampling location but are unclustered. ARDs are clustered along rows using Euclidean distances with complete linkage. ARD names by row can be viewed in the source data for **Figure 2**. Color scale values indicate the number of normalized alignments per ARD per sample. (**B**) Histogram of unique ARDs identified per sample (N=87). See **Figure 2—source data 1** for raw count matrix of ARDs by sample, which was used to produce heatmap and histogram.

The following source data is available for figure 2:

**Source data 1.** Raw and normalized count matrix of ARDs (rows) identified by sample (columns).

within 17 drug classes (*Supplementary file 3*). Reads aligning to genes that encode resistance to tetracyclines and the macrolide-lincosamide-streptogramin classes of antimicrobials were most abundant, with the TetQ, TetW, TetO, and mefA gene families comprising the majority of alignments within these two classes (see Source Data for *Figure 2A*).

To assess systematic changes in resistome composition during the feeding period (i.e., from arrival to exit, truck and holding samples), non-metric multidimensional scaling (NMDS) ordination at the ARD level was performed using Hellinger transformation and Euclidean distances to avoid over-weighting of rare ARDs, which were prevalent in these data (*Figure 2A*) (*Legendre and Gallagher, 2001*). Samples with only one ARD (n = 2) were removed from clustering analyses. Pre-slaughter samples clustered by sample matrix (i.e., feces, soil, water and swabs, the latter of which comprised all truck samples) based on ARD composition (*Figure 3A*), suggesting that the resistomes in these sample types differed significantly. Therefore, to avoid confounding, we performed ordination separately on fecal, soil and water samples, all of which exhibited a significant shift from arrival to exit or holding (*Figure 3B–D*); the truck resistome could not be compared owing to complete confounding between sampling location and matrix type. While this shift could result from AMD exposures, culture-based studies of phenotypic resistance have reported mixed results when investigating potential associations between feedlot AMD use and AMR (*Rao et al., 2010*; *Morley et al., 2011*). In addition to AMD exposures, cattle undergo numerous changes during the feeding period, including maturation and a gradual shift from forage-based to high-energy rations, all of which have been shown to affect the fecal microbiome in swine, although little is known about these factors in beef cattle (*Umu et al., 2015*; *Mach et al., 2015*; *Kim et al., 2012*). Therefore, changes in the resistome could also be driven by changes in bacterial community composition, a phenomenon recently reported for a set of functionally confirmed metagenomic soil samples (*Forsberg et al., 2014*). In order to measure correlation between the microbiome and resistome, and thus to investigate the extent to which microbiome changes may have been driving the resistome, we performed post-hoc procrustes analysis, which can be used to determine the degree of correlation between two ordinations (*Peres-Neto and Jackson, 2001*). This analysis confirmed a high correlation between the resistome (ARD level) and the microbiome (species level) in arrival and exit samples (*Figure 4A–B*). However, there was tighter correlation on arrival than on exit, leading to the hypothesis that additional factors such as AMD exposures may have influenced resistome changes independently of the microbiome.

Resistome richness at the ARD level decreased significantly during the feeding period (Wilcoxon paired signed-rank p=0.002). Interestingly, this decrease occurred primarily through the loss of resistance to AMDs that were not used on cattle in this study population or in the feeding systems in which they were raised (e.g., phenicols, aminocoumarins, elfamycins, rifampin, bacitracin and polymyxin B, *Figure 5*). In contrast, resistance to macrolides and tetracyclines remained prevalent in exit and holding pen samples, and these were the two classes of AMDs most commonly administered to cattle within study pens (*Table 1*). In addition, 5 of the 8 pens contained several cattle that received fluoroquinolones, and 3 of the 8 contained several cattle that received β-lactam AMDs. One hypothesis for the finding of reduced ARD richness is that the pattern of antimicrobial exposures during the feeding period created pressure on microbes that favored maintenance of ARDs that conferred a fitness advantage (i.e., ARDs protecting against macrolides and tetracyclines). Conversely, ARDs conferring resistance to antimicrobials that were not used in this study population (e.g., phenicols, aminocoumarins, elfamycins) or non-drug-specific ARDs (e.g., porin modification genes) did not confer a competitive advantage and may have exacted a fitness cost to bacteria carrying them (*Vogwill and MacLean, 2015*). Thus, such bacteria may have gradually diminished in the population, leading to disappearance of these ARDs within the pan-microbial population and thus an observed decrease in ARD richness. These pan-microbial population genetic changes may also explain the apparent decoupling of the resistome and the microbiome that was observed from feedlot arrival to exit (*Figures 4A–B*). However, more research is needed to understand how selective pressures acting on specific bacteria may manifest at the pan-microbial level.

Interestingly, the aminoglycoside class was an exception to this pattern, as aminoglycoside resistance remained prevalent throughout the feeding period despite the absence of these drugs in this study population and in beef production in general. While this study was not designed to definitely test for AMD-ARD associations, these findings suggest that the relationship between AMD use and AMR is not straightforward. In addition, we identified several ARDs that confer resistance to critically

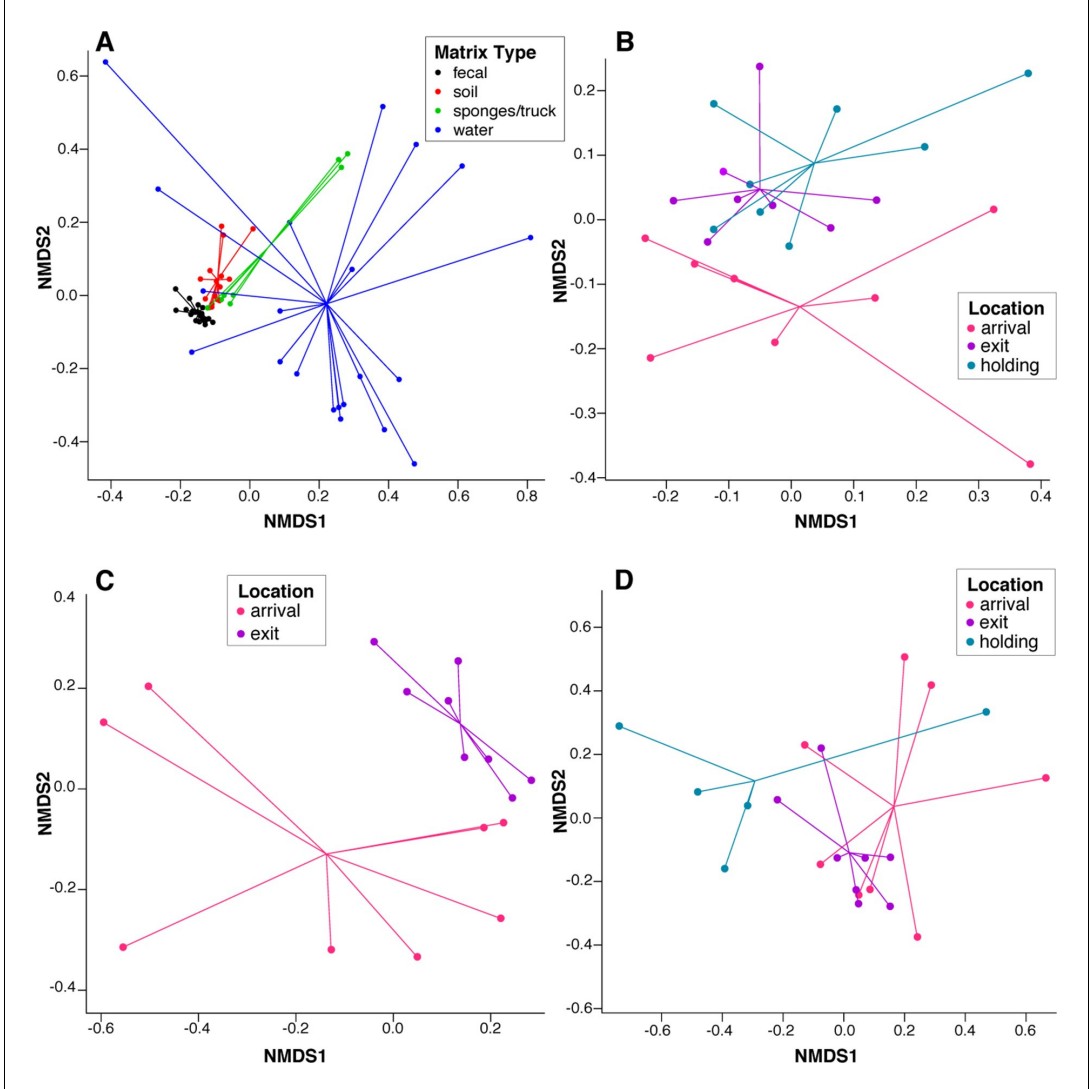

**Figure 3.** NMDS ordination plots of ARD composition, by sample type and location. Non-metric multidimensional scaling (NMDS) ordination plots of pre-slaughter sample ARD composition, depicting significant sample separation by (**A**) matrix (Stress=0.13, R=0.41, p=0.001), and location within (**B**) feces (Stress = 0.10, R=0.03, p=0.04), (**C**) soil (Stress = 0.05, R=0.34, p=0.006) and (**D**) water (Stress=0.10, M=0.29, p=0.005).

important antimicrobial drugs in humans when expressed in disease-causing agents (*Table 2*); however, these AMDs were not used in this study population and most are not sold for use in cattle. An arrival soil sample contained the functionally-confirmed carbapenemase (bla)OXA-235 (*Higgins et al., 2013*) as well as vgaD and vatG ARDs, which together confer resistance to quinupristin-dalfopristin (*Jung et al., 2010*). Another arrival soil sample and a water sample from a holding pen contained (bla)OXA-347, which has only been shown to confer resistance to ampicillin but is classified as a carbapenemase based on 53% amino acid identity (*Cheng et al., 2012*); therefore, it is unknown whether this is a true carbapenemase ARD. Of the 7 sequenced water samples collected from holding pens, 3 contained reads aligning to the strict carbapenemase class bla(cphA). Additionally, the 4 truck samples collected in Texas all contained a CfrA 23S rRNA methyltransferase, which confers resistance to phenicols, lincosamides, oxazolidinones (linezolid), pleuromutilins, and streptogramin A (PhLOPS$_A$). Despite the presence of these ARDs, study cattle were not exposed to aminoglycosides, carbapenems, streptogramins, phenicols, lincosamides, linezolid, or pleuromutilins while in the feedlot (*Table 1*). In addition, carbapenems, pleuromutilins and linezolid are not approved for use in cattle production, and therefore antimicrobial use practices cannot directly

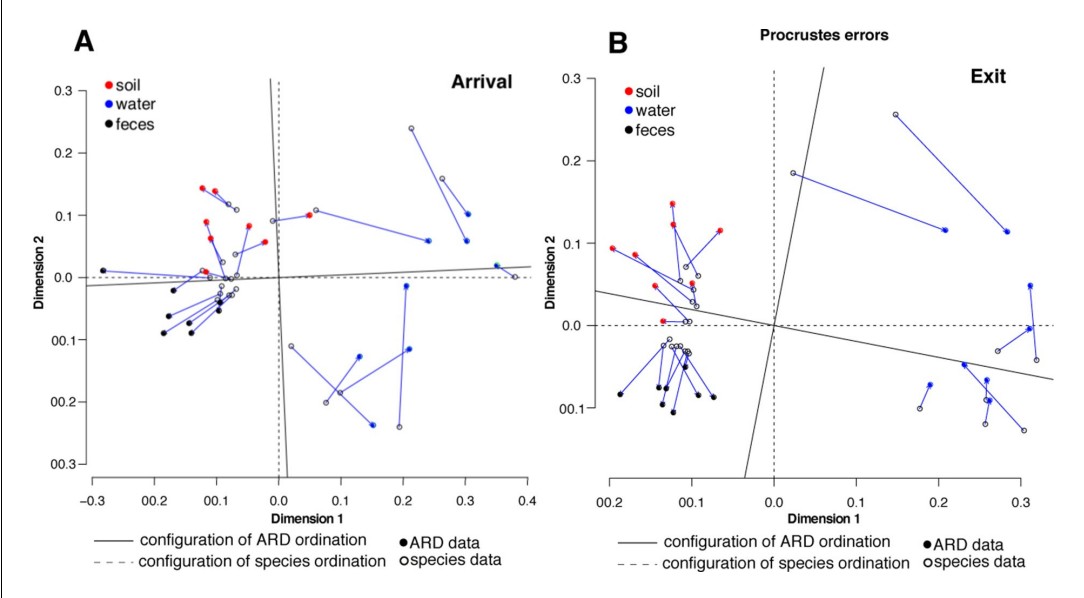

**Figure 4.** Procrustes analysis of ARD content (filled circles) and species composition (open circles) at arrival (A) and exit (B) using Hellinger transformation and NMDS ordination (*Legendre and Gallagher, 2001*). Stress values for ARD ordination at arrival and exit were 0.08 and 0.03, respectively, and for microbiome species ordination at arrival and exit were 0.06 and 0.07, respectively. Soil (red), water (blue) and fecal (black) samples clustered significantly in the microbiome and resistome data. Procrustes configurations were correlated in the arrival and exit samples, but less so in the exit samples ($M^2$ = 0.29 and 0.18, respectively).

explain the presence of these important ARDs. We also identified bla(TEM)-116, an extended-spectrum β-lactamase (ESBL) ARD, in 1 water holding pen sample. Third-generation cephalosporins (a class of β-lactams) are used in beef cattle production, although other drugs are often used more commonly as reflected in this study (*Table 1*). We did not identify ARDs from the bla(SHV), bla(CTX-M) or bla(CMY) classes of ESBLs, or the carbapenemases bla(SME), bla(IMI), bla(NDM), bla(GES) or bla(KPC) (*Table 2*).

It is important to note that these ARDs were present in extremely low relative abundance, with fewer than a dozen of the 1.2 M ARD-assigned reads aligning to each (*Figure 2—source data 1*). While high sequence homology and the resistance-conferring functional residues were intact (*Table 2*), we cannot infer phenotypic expression from these data. Furthermore, while this is the first published report of these ARDs in the feedlot setting, it is also the first study to utilize a shotgun metagenomics approach on feedlot samples; therefore, we cannot contextualize these findings with respect to previous research and we cannot determine whether presence of these ARDs in feedlot samples is a novel or long-standing phenomenon. Identifying these ARDs in metagenomic data may provide important insight above and beyond a culture- or PCR-based approach, but additional work is needed to understand the biological, ecological, and public health consequences of these findings (*Martínez et al., 2015*). For instance, the presence of these ARDs in the feedlot soil could be explained by transfer into the feedlot environment through either other cattle or fomites (e.g., feedlot workers, feedlot working dogs and horses), or through air or water. Use of other AMD classes could also co-select for these ARDs within the cattle population. These findings highlight the complexity of the AMD–AMR relationship, as well as the fact that food production is intrinsically linked to other ecosystems via diffuse environmental contacts. Given these complexities, we believe an ecological and metagenomic approach is necessary to thoroughly and comprehensively research this important public and human health issue.

ARD composition did not differ between pens of cattle (n=8) or feedlots (n=4) when ordinated using NMDS (*Figure 6*). This was notable given the geographic separation of the 4 feedlots, the fact that the pens did not have contact with one another, and the variability in parenteral AMD exposures across the 8 study pens (*Table 1*). However, common management strategies used in all 4 feedlots and the ubiquitous use of in-feed macrolides within study pens could explain this lack of

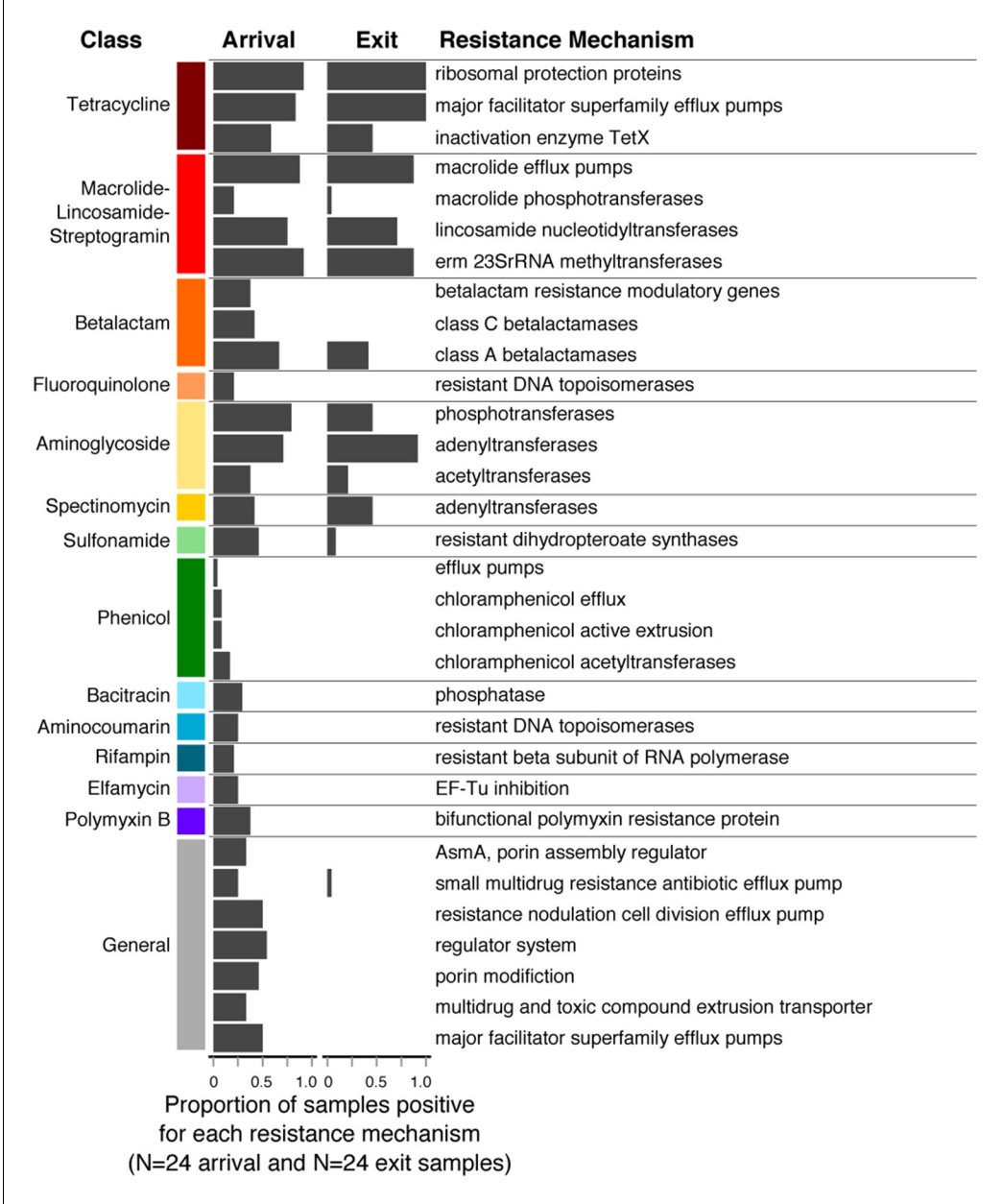

**Figure 5.** Changes in prevalence of resistance mechanisms during the feedlot period (arrival to exit). Proportion of arrival (n=8 soil, 8 fecal, 8 water) and exit (n=8 soil, 8 fecal, 8 water) samples that contained at least one ARD in each resistance mechanism (n=33), grouped by resistance class.

difference. In addition, water runoff, windborne dust and fomites within feedlots could contribute to a mixing of pen resistomes, despite differential AMD exposures across pens. Resistome composition did show statistical difference by geography at the ARD level (i.e., Colorado versus Texas feedlot, truck, and holding pen samples), but not at the level of resistance mechanism or class. Despite statistical significance, both the $R^2$ value from adonis results and the R-statistic from anosim results suggest that the biological relevance of the differences between states was not high (see *Figure 6— source data 1*). These findings support the idea of environmental connectivity within feedlots, and perhaps within and between geographic regions. However, further study is needed to determine what effect common management practices may have had on these results.

**Table 2.** ARDs to critically important antimicrobials that were specifically searched for in all 87 samples.

| Classification | Group | ARD ID (database) | Sample Type, Location | Alignment characteristics |
|---|---|---|---|---|
| Carbapenemases | bla (OXA) | Bla)OXA-347:JN086160 (ARG-ANNOT)[a] (Bla)OXA-235:JQ820240 (ARG-ANNOT) | 1 x water, holding 1 x soil, arrival 1 x soil, arrival | 100% nucleotide homology with reference across ≥ 80% of gene (≥ 1x coverage) 100% nucleotide homology to KSG, FGN and STFK motifs |
| | bla (SME) | Not identified | Not identified | Not identified |
| | bla(IMI) | Not identified | Not identified | Not identified |
| | bla (NDM) | Not identified | Not identified | Not identified |
| | bla (GES) | Not identified | Not identified | Not identified |
| | bla (KPC) | Not identified | Not identified | Not identified |
| | bla (cphA)[b] | cphA1_2_AYAY261377 (Resfinder) | 3 x water, holding | 100% amino acid homology to reference (4 silent substitutions) |
| Extended-spectrum β-lactamase | bla (TEM) | (Bla)TEM-116:AY425988 (ARG-ANNOT) | 1 x water, holding | 100% nucleotide homology with reference across ≥ 80% of gene (≥ 1x coverage) |
| | bla (SHV) | Not identified | Not identified | Not identified |
| | bla (CTX-M) | Not identified | Not identified | Not identified |
| | bla (CMY) | Not identified | Not identified | Not identified |
| Quinupristin-dalfopristin resistance | vga/vat | GQ205627.2.gene3 (CARD) AND (MLS)VgaD:GQ205627:1394-2971:1578 | 1 x soil, arrival | 100% nucleotide homology to Walker A and B motifs; silent substitution in RSGG motif 100% nucleotide homology to LβH hexapeptide repeat domain |
| Multi-drug resistance to PhLOPS$_A$ [c] | cfr | (MLS)CfrA:AM408573 (ARG-ANNOT) | 4 x swab, truck (all in Texas) | 100% nucleotide homology with reference across ≥ 95% of gene (≥ 1x coverage) |

[a]Note that phenotypic resistance to carbapenems has not been confirmed for this ARD

[b]bla(cphA) is a strict carbapenemase

[c]Confers multi-drug resistance to phenicol, lincosamide, oxazolidinones (linezolid), pleuromutilins, and streptogramin A

When examining the 16 post-slaughter samples obtained from the belts, tables, and meat trimmings, no ARDs were identified in any of these market-ready samples (n = 8 pooled belt/table samples and n=8 trimming rinses). These samples yielded large amounts of DNA, but >99% of the reads aligned to the bovine genome (*Supplementary file 1*); therefore, the lack of detection of ARDs could likely be attributable to low-sequencing coverage of bacterial DNA. However, there are also plausible biological explanations for the lack of bacterial DNA (and thus ARDs) in these samples. The bacterial contamination of beef during slaughter occurs primarily during the removal of the hide and gastrointestinal tract (GIT), at which point the surfaces of carcasses can routinely be contaminated with aerobic bacterial counts of 6.1 – 9.1 $\log_{10}$ CFU/100 cm$^2$ (*Bacon et al., 2000*). To decrease this contamination, it is standard in North America for carcasses to undergo several highly effective antibacterial interventions after hide and GIT removal, including steam vacuuming, carcass washing, application of organic acid rinses and thermal pasteurization (*Bacon et al., 2000*). All carcasses in this study underwent each of these interventions sequentially, a process that has been shown to reduce bacterial loads by >5 $\log_{10}$ CFU/100 cm$^2$ total plate count (*Greig et al., 2012*). Nevertheless, nationwide food safety surveys in the U.S. suggest that a relatively low level of bacterial presence is common on post-slaughter beef trimmings, with aerobic plate counts routinely containing $10^2$–$10^3$ CFU/g of trimmings (*USDA, FSIS, 2011*). It is likely that some unknown proportion of these

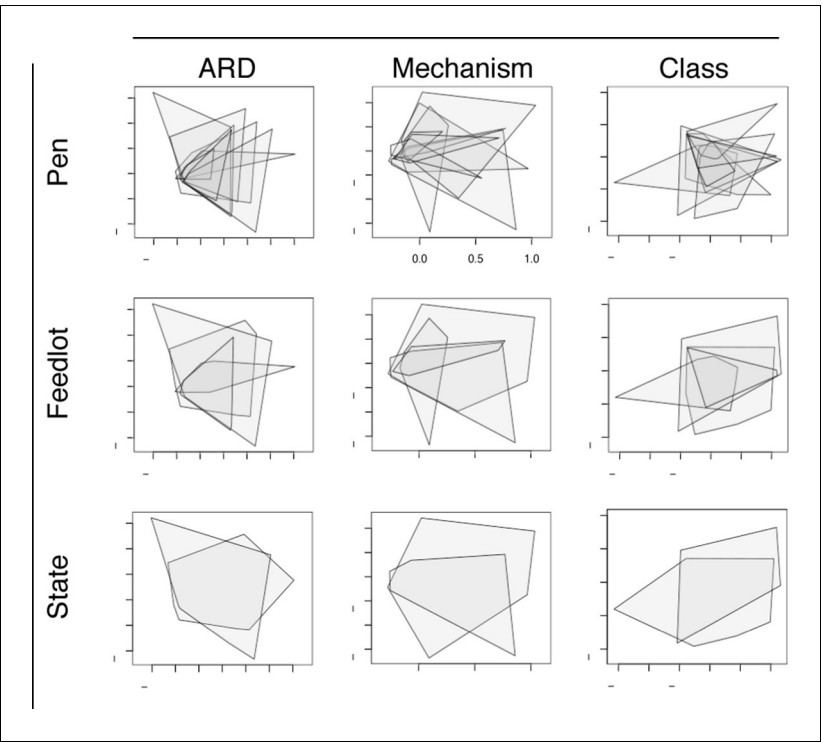

**Figure 6.** Non-metric multidimensional scaling (NMDS) ordination plots at the ARD, mechanism and class levels, visualized by pens (n=8), feedlots (n=4) and states (n=2). In each NMDS plot, a polygon corresponds to one unit (i. e., pen, feedlot or state) and represents the convex hull for that unit (i.e. the smallest amount of space within the graph that contains all points (or samples) within that unit). To view results of NMDS ordination, as well as adonis and anosim statistics, see *Figure 6—source data 1*.

The following source data is available for figure 6:

**Source data 1.** NMDS ordination, adonis, and anosim results at the ARD, mechanism and class levels, by pen, feedlot, and state variables.

organisms harbor ARDs, suggesting that the shotgun metagenomics assay as used in these high-host background samples may have had low sensitivity to detect these very low-abundance ARDs.

To differentiate between the effects of incomplete sequencing and true reduction of the microbial population, we conducted a microbiome analysis of samples collected pre- and post-slaughter, the former comprising all arrival, exit, truck, and holding pen samples. We found that microbiome (i. e., total microbial community) diversity was reduced in the post-slaughter samples compared to the pre-slaughter samples, suggesting a dramatic alteration in the composition of the microbial community, which could reflect the impact of antibacterial interventions that are applied during slaughter (*Figure 7A*). However, low diversity could also be the result of low-coverage sequencing of microbial DNA in the post-slaughter samples (*Smith et al., 2014*). Therefore, we conducted a closer analysis of differential microbial abundance between pre- and post-slaughter samples using zero-inflated Gaussian mixture models to account for distinct processes for zero-count data (i.e., true absence vs. incomplete sequencing depth) (*Paulson et al., 2013*). Pairwise comparisons between pre- and post-slaughter samples were performed using limma's makeContrasts function (*Smyth, 2004*), and pen ID was added to all models as a covariable to account for repeated measurements. We identified 416 (of 763) differentially abundant genera and 840 (of 1821) differentially abundant species, the vast majority of which were more abundant in the pre-slaughter samples (*Figure 7B*). Of the 19 genera and 68 species that were more abundant in post-slaughter samples, many are known to be heat-tolerant and/or environmentally hardy bacteria, archaea, viruses, and phages (see *Figure 7—source data 1*); e.g., *Staphylothermus*, *Pyrococcus*, *Pseudomonas* and *Pleurocapsa*, suggesting that they were able to withstand the high heat and other harsh conditions utilized as part of the slaughter

antimicrobial interventions. However, sample matrices in the pre- and post-slaughter samples were confounded with location. Therefore, we performed the same analysis solely on swab samples (pre-slaughter swabs from the transport trucks and post-slaughter swabs from the abattoir conveyor belts). Interestingly, the same overall pattern of change was observed; the majority of bacterial genera (105/757) exhibited decreased abundance post-slaughter (see *Figure 7—figure supplement 1*). However, unlike the comparison across all matrices, the genera with increased relative abundance in post-slaughter swab samples were not characterized by a large proportion of environmentally hardy bacteria ( *Figure 7—source data 1*). Given that the truck environment is also relatively harsh (i.e., extreme temperatures, lack of nutrients, dry conditions), this result was not unexpected. Taken together, these results suggest that the multiple-hurdle interventions decrease overall bacterial load, and thus greatly diminish the abundance of ARDs in post-slaughter beef. However, the incomplete sequencing depth in the post-slaughter samples cannot be fully discounted and future metagenomic investigations of samples with high-host background should consider methods to overcome this limitation. We were unable to compare these results to previous studies in the abattoir setting, as most of these bacteria are difficult or impossible to culture and therefore have not been investigated. Interestingly, culture-based resistance surveillance in beef trimmings reported >30% prevalence of resistance amongst *Salmonella* isolates (**USDA, FSIS, 2011**). However, *Salmonella* was recovered from <1% of these trimming samples (22 of 1,791 samples collected in 2011), suggesting that the recovery of resistant *Salmonella* is a very rare event; in this respect, the results of these culture-based efforts concur with our findings. Furthermore, samples taken as part of the culture-based surveillance program were enriched prior to isolation, likely resulting in increased sensitivity for

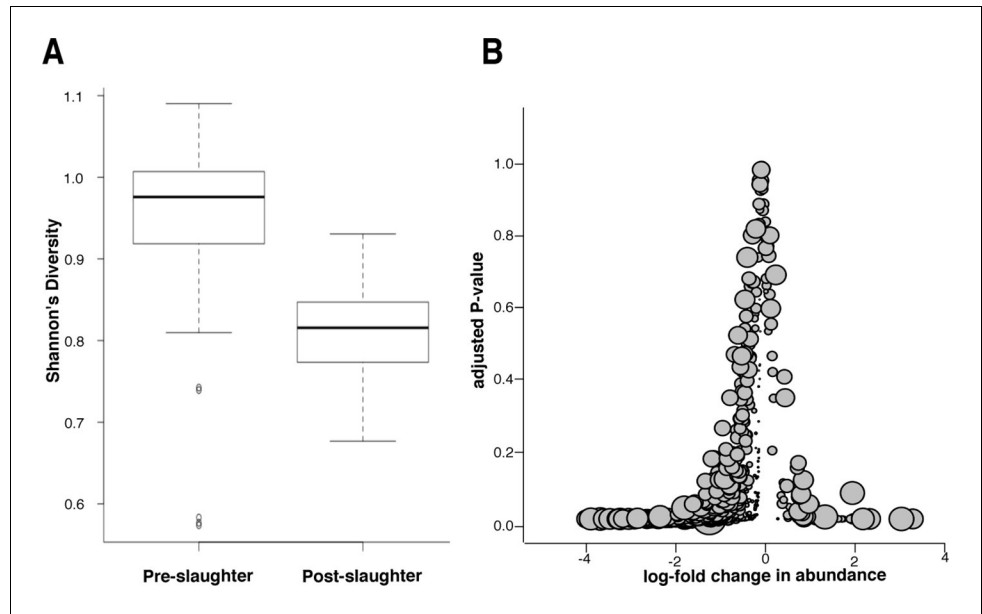

**Figure 7.** Microbiome changes from pre- to post-slaughter in all samples. (**A**) Boxplot of Shannon's diversity at the species level, pre- vs. post-slaughter across all sample matrices. Shannon's diversity was significantly lower in post-slaughter samples when tested using Wilcoxon paired rank test (*P*<0.0001). See source data for *Figure 7* (sheet '*Figure 7A*') for Shannon's Diversity Index by sample, which was used to produce boxplots. (**B**) Log$_2$-fold change in abundance of genera from pre- to post-slaughter versus adjusted *P*–value, across all samples matrices. Dot size is proportional to the average abundance of the genus across all samples. For taxa table and counts used to produce model of log$_2$-fold change in abundance, see *Figure 7—source data 1* (sheet '*Figure 7B* taxa table'); for model output, see *Figure 7—source data 1* (sheet '*Figure 7B* model output').

The following source data and figure supplement are available for figure 7:

**Source data 1.** Shannon's diversity, taxa table and model output for *Figure 7*.

**Figure supplement 1.** Microbiome changes from pre- to post-slaughter in swab samples only.

*Salmonella* detection compared to the shotgun metagenomics approach. Further work is needed to compare the sensitivity and specificity of shotgun metagenomics with more traditional assays.

## Conclusion

This study reports the use of shotgun metagenomics in a novel investigation of AMR that tracked specific pens of intensively-managed cattle from feedlot entry through slaughter to market-ready product in a longitudinal fashion. While our results are more directly relevant to large North American feedlot operations, the general approach can easily be extended to other sectors of beef production, other countries, and other livestock production systems. This is particularly important given the increasing availability of next-generation sequencing machines worldwide. However, our study also emphasizes the need to properly contextualize shotgun metagenomics results, including understanding the limitations of sequencing coverage in different matrices.

While our results suggest that slaughter-based intervention systems minimize the likelihood of intact ARDs being passed through the food chain, they also highlight the potential risk posed by indirect environmental exposures to the feedlot resistome. This concern is especially salient given evidence in this study that ARDs may be 'shared' between the pens of cattle and feedlots within a geographic region, indicating environmental connectivity that could also extend to human habitats through wastewater run-off, manure application on cropland, and windborne particulate matter. The pattern of resistome change during the feeding period suggests that AMD use practices may be a driving force shaping the feedlot resistome, but more research is warranted. In particular, this study utilized pooled samples to explore resistome dynamics within and between groups of animals; the excrement or meat products from animals treated with a rarely used antimicrobial (e.g., a fluoroquinolone or β-lactam) were less likely to be included in the pooled samples compared to end-products of animals treated with a commonly-used antimicrobial (e.g. in-feed macrolide). Future research should include the sampling of individual treated and untreated animals to better discern the effect of AMD exposure on the resistome. Furthermore, the scientific community urgently needs to develop a better understanding of the risk of different resistomes and resistance genes (*Martínez et al., 2015*) and to unify ARD nomenclature so that databases are standardized and analyses are comparable across studies (*Hall and Schwarz, 2015*). Finally, this study highlights the utility of an ecological, metagenomics, and systems approach to investigating AMR in food production, and provides unique insights that can be used to better inform agricultural and public health policy.

## Materials and methods

### Antimicrobial use data

AMU data were obtained from feedlot owners and/or managers, and were aggregated and analyzed at the pen level because the samples were collected as pooled composites.

### Sample size estimation and sampling design

Due to a lack of published studies on the resistome and/or microbiome of beef feedlots and abattoirs, we were unable to perform a formal sample size estimation. Somewhat related studies have utilized <5 animals and <3 biological replicates per animal (*Wichmann et al., 2014*; *Durso et al., 2011*), and therefore we felt that composite samples from 8 pens would provide a more representative sample set than those used in currently published studies. While it is unlikely that pooled samples represent all individual animals within a pen or unit, they do represent a group-level sample that can be used to understand group-level dynamics. This is especially true given research indicating the tendency for cohabiting individuals to share similar microbiomes (*Song et al., 2013*; *Yatsunenko et al., 2012*). Because livestock animals are typically managed in groups, pooled samples are likely the appropriate unit of sampling. In addition, recent work has shown that pooled samples do not produce statistically or biologically significant differences in prevalence estimates for resistant generic *Escherichia coli* compared to individual samples (*Benedict et al., 2013*). In order to balance external representativeness with biological replicates, 4 feedlots and 2 pens per feedlot were selected for sampling. Sample types ('matrix') were chosen based on relevance to public health risk (composite feces, soil, water in the feedlots; internal truck walls during transport; feces and water in holding pens; and end of conveyor belts and trimmings in the abattoir). Selected sampling

time points (i.e., arrival, exit, transport/truck, holding, and post-slaughter) represented points in the process of beef production when ARD abundance or transmission may be influenced by production practices, and interventions can be optimally assessed or implemented.

## Sample collection

Fecal, soil, swab, and trimming samples were collected using sterile gloves sprayed with alcohol and placed into sterilized Whirl-Pak bags (Nasco, Fort Atkinson, Wisconsin). Water samples were collected into containers that had been submerged in bleach for 5 min, rinsed with sterile water and autoclaved. Pooled fecal samples were collected from feedlot and holding pen floors; pooled soil samples were also collected from feedlot pen floors, but not holding pen floors, which were concrete. Investigators walked through pens on diagonal lines, collecting ~30 g of feces or soil from 12 equally spaced locations. The 12 soil and 12 fecal samples from each pen were then placed in one Whirl-Pak bag each (Nasco) and mixed thoroughly. Water trough contents in each pen were thoroughly mixed, and water samples (1 L each) were collected and placed into sterile containers. Truck samples were collected using an EZ Reach polyurethane sponge pre-hydrated with 10 mL Dey/Engle neutralizing broth (World Bioproducts LLC, Mundelein, Illinois), which was used to swab the internal walls of each truck (sides, door and floor) 20 times on the front and back of each sponge. For each pen, 3 of 5 trucks were randomly selected for sampling, and 1 sponge was used per truck. After slaughter, carcasses in each pen were grouped and processed by USDA quality grade. Post-slaughter samples were obtained when the USDA grading group with the greatest number of carcasses was being disassembled and processed. In the slaughter room, EZ Reach polyurethane sponges pre-hydrated with 10 mL Dey/Engle neutralizing broth were used to collect swab samples at the end of conveyor belts used to process chuck (i.e., the shoulder) and round (i.e., hind leg) primal cuts, and trimmings. The end of these belts represented the last stage in the slaughter and disassembly process, immediately prior to beef being packaged for retail distribution. Sponges were held on each running belt for one minute per side. Beef trimming samples were collected from the trim conveyor belt immediately prior to spraying of the last antimicrobial solution in the slaughter process.

All samples were transferred on ice to the Center for Meat Safety & Quality at Colorado State University. The samples collected in Colorado arrived within one hour of collection, and the samples collected in Texas arrived within 48 hr. Upon arrival, fecal, soil, swab, and trimming samples were immediately frozen at -80°C. Water samples were centrifuged at 15,000x$g$ for 20 min at 4°C, and 5 mL of the pellet was collected for DNA extraction.

## Sample processing

### Pre-extraction

All samples were thawed prior to DNA extraction. A sedimentation step was used to process the feces and soil prior to DNA extraction, allowing for the simultaneous sedimentation of heavier soil/fecal debris and the release of bacterial cells into the upper supernatant. This step made it possible to process a greater volume of sample matrix (up to 10 g) while removing additional PCR inhibitors known to be present in soil and feces (*Tsai and Olson, 1992*), resulting in a more complete representation of bacterial DNA presence. Briefly, 30 mL of BPW was added to 10 g soil or feces in a 50 mL conical tube, and the samples were shaken vigorously to mix well before being allowed to sediment on the bench for 10 min. Supernatants, including limited soil/fecal debris, were transferred to a new 50 mL conical tube and centrifuged for 10 min at 4300x$g$. The BPW was removed, and the resulting sample pellet was rinsed with 5 mL of molecular grade sterile PBS and centrifuged again at 4300x$g$ for 10 min. The supernatant was removed, and the resulting pellet was re-suspended in 15 mL of PowerBead solution before being transferred to the PowerMax Bead Solution Tube provided in the kit and proceeding with the DNA extraction protocol.

Four hundred grams of thawed meat from each trimming sample were rinsed in 90 mL of buffered peptone water (BPW) and then placed at 4°C to separate and harden the lipid content. After hardening, the liquid portion of the sample was removed and centrifuged at 4280x$g$ at 4°C for 20 min, after which the supernatant was discarded and the pellet re-suspended in 5 mL of cold, sterile saline. The cold saline wash was repeated, and after the final centrifugation, 250 mg of the resulting pellet was utilized for extraction. Sponges used to collect swab samples were squeezed with a Brayer squeegee to remove the broth liquid. The sponges were then rinsed in 10 mL of BPW and

squeegeed again. The rinsate from both rounds of squeezing were then centrifuged at 4300x*g* at 4°C for 20 min, the supernatant was removed, and the pellets were re-suspended in phosphate buffered saline (PBS), at which point the pellets from samples collected from the same pen were combined, resulting in 1 pooled truck sample and 1 pooled belt sample per pen. The combined samples were then centrifuged again at 4280x*g* at 4°C for 20 min, and 250 mg of the resulting pellet was weighed and set aside for DNA extraction.

## DNA extraction

The Mo Bio PowerMax Soil DNA Isolation Kit was used to extract DNA from 10 g/sample of pooled feces and soil, whereas the Mo Bio PowerSoil DNA Isolation Kit was used to extract DNA from 250 mg/sample of water, swab, and trimming pellets (Mo Bio Laboratories, Inc., Carlsbad, California). Different kits were used to accommodate different sample volumes (as recommended by the manufacturer); however, they utilize identical reagents and chemistries. DNA extraction was performed according to the manufacturer's protocol. DNA for fecal and soil samples was eluted in 5 mL of the kit elution buffer, and water, swab, and trimming rinsate samples were eluted in 50 µl of the kit elution buffer to maximize DNA concentration. After extraction, DNA concentration was measured at 260 nm using a NanoDrop spectrophotometer (Thermo Fisher Scientific, Inc., Waltham, Massachusetts). The samples that did not have a concentration of at least 20 ng/µl (1 µg total in 50 µl) were precipitated using a traditional ethanol precipitation procedure. To the final DNA sample, 1/10 volume of 3 M sodium acetate, pH 5.2, was added. Two volumes of cold 100% molecular grade ethanol was added, and the sample was mixed several times by inversion before incubating at -20°C for 1 hr. The samples were centrifuged at 11,000 x g for 20 min at 4°C. Supernatants were carefully discarded and 150 µL 70% cold ethanol was added and mixed by inversion. The samples were centrifuged a final time at 11,000 x g for 10 min at 4°C. Supernatants were again discarded and the DNA pellets allowed to air dry before resuspending in ¼ the original DNA volume with Solution C6 included in the Mo Bio DNA extraction kits.

## Sequencing

After DNA extraction or concentration, 100 µl of each fecal and soil DNA and 30 µl of each water, swab, and trimming rinsate DNA were delivered on ice to the Genomics and Microarray Core at the University of Colorado Denver. Libraries were constructed using the Illumina TruSeq DNA Library Kit (Illumina, Inc., San Diego, California ) for samples that contained at least 1 µg of DNA and using the NuGEN Ultra Low DNA Library Preparation (NuGEN Technologies Inc., San Carlos, California) for samples that contained less than 1 µg of DNA, following the manufacturer's protocols. Paired-end sequencing was performed on the Illumina HiSeq 2000 (Illumina, Inc.).

## Bioinformatics

### Creation of master, non-redundant ARD database

Resfinder (*Zankari et al., 2012*), ARG-ANNOT (*Gupta et al., 2014*) and CARD (*McArthur et al., 2013*) databases were chosen for the foundation of the master database because they are specific to antimicrobial resistance genes, are actively curated and frequently updated; all 3 databases were downloaded on August 12, 2014. Redundant sequences between ARG-ANNOT and Resfinder were identified using CD-HIT-EST-2D (*Fu et al., 2012*) with local alignment (-G 0) and the following parameters: -c 1.0 -AS 0 -AL 0 -aL 1.0 -aS 1.0. A single representative sequence was selected from each resulting cluster (n=1,427), and these sequences were appended to the list of unique gene sequences in ARG-ANNOT (n=261) and Resfinder (n=715). This process was then repeated for the CARD database using the combined ARG-ANNOT/Resfinder non-redundant database. Seven hundred and eight sequences were unique to CARD, resulting in a final non-redundant database containing 3111 unique ARD sequences.

### Bioinformatics pipeline used to identify ARDs

Raw sequence data were obtained from the Genomics and Microarray Core at the University of Colorado Denver. Reads were filtered for quality using Trimmomatic (*Bolger et al., 2014*) in the following manner: first, the leading 3 and trailing 3 nucleotides were removed from each read, then a sliding window of 4 nucleotides was used to remove nucleotides from the 3' end until the average

Phred score across the window was at least 15. Trimmomatic's 'ILLUMINACLIP' command was used to remove adapters supplied in the TruSeq3 adapter sequence file. A maximum of 2 mismatches were allowed in the initial seed, and adapter clipping occurred if a match score of 30 was reached. In addition, both reads were retained upon clipping, despite probable complete sequence redundancy, to supply more reads for downstream applications.

After clipping and trimming, reads were matched to the *Bos taurus* reference genome (UMD_3.1) using Kraken (*Wood and Salzberg, 2014*) in 'quick operation' mode; reads with <5 31-mers matching to the *Bos taurus* genome ('non-host' reads) were extracted for further analysis. Non-host reads were then aligned to the master, non-redundant ARD database using BWA with default settings (*Li and Durbin, 2009*). A custom-developed Java-based script was used to parse the resulting SAM file such that the gene fraction was calculated for each ARD identified in each sample; this can be accessed at https://github.com/colostatemeg/gene_fraction_script/releases. Gene fraction was defined as the proportion of nucleotides in the ARD that aligned with at least one read. In order to decrease the number of false positive ARD identifications (*Gibson et al., 2015*), only ARDs with gene fraction of >80% were defined as present in the sample and included in further analyses.

Each identified ARD was classified at the mechanism and class levels (*Supplementary file 3*). For each ARD in each sample, the total number of aligned reads was summed to create a count matrix with samples in columns and ARDs in rows. ARDs present in fewer than 3 samples were removed from further analysis due to an inability to accurately normalize such counts (N = 83 out of 319 ARDs). Counts for remaining ARDs were normalized using cumulative sum scaling (*McMurdie and Holmes, 2014*). Due to the sparseness of count data, a default percentile of 0.5 was chosen for normalization, based on published recommendations (*Paulson et al., 2013*). Finally, normalized counts were aggregated to the mechanism and class levels. These count matrices were used for ordination and heatmap generation.

## Statistical analysis

Pre-planned analyses included statistical testing of resistome NMDS ordination results by matrix type, sampling location, pen, feedlot and state; formal statistical comparison of resistome richness and diversity metrics between the same factors; and multivariable modeling of $log_2$-fold change in abundance of ARDs, resistance classes and mechanisms between sampling locations. Additional analyses, including those related to the microbiome and procrustes rotation of NMDS ordination results, were performed post-hoc for purposes of hypothesis generation.

## Ordination and heatmap generation

All ordinations were pre-planned and were conducted on 2 dimensions with 'vegan's' metaMDS function (*Oksanen et al., 2014*), using Euclidean distances between Hellinger transformed read counts that had been normalized using CSS, as described above (*Legendre and Gallagher, 2001*). The metaMDS function enables the discovery of a stable ordination solution using many random starts. The significance of study variables in explaining ordination variation was tested using permutational multivariate analysis of variance using distance matrices as implemented in 'vegan' (function 'adonis') (*Anderson, 2001*). In addition, we included post-hoc statistical tests of NMDS ordinations using the Analysis of Similarity test as implemented in 'vegan' (function 'anosim'), in order to provide a measure of effect with the corresponding R-statistic. Post-hoc procrustes superimposition was performed on results of NMDS ordination of resistome ARD and microbiome species composition, for arrival and exit samples, and the $M^2$ statistic was used to assess correlation of ordinations. A non-column-clustered heatmap (*Figure 2A*) was generated on counts of ARDs that had been normalized using CSS as described above. Rows were clustered using the complete linkage method.

## Richness and diversity comparisons

Richness was defined as the number of unique features (ARDs, mechanisms, classes, species or genera) in a sample, while diversity was calculated using Shannon's Index. Pre-planned comparisons of richness and diversity between samples were conducted using paired Wilcoxon signed rank test due to the presence of repeated measures when comparing different sampling locations (e.g., arrival vs. exit, pre- vs. post-slaughter).

## Microbiome classification

Microbiome analysis was conducted post-hoc as a means to identify potential shifts in the microbial community as a result of pathogen-reduction interventions during the slaughter process; and to identify the amount of correlation between the microbiome and resistome during the time cattle were in the feedlot. Kraken was used to classify reads phylogenetically, using default settings (*Wood and Salzberg, 2014*). A very high number of reads for all samples were assigned to *Achromobacter xylosoxidans* strain NBRC 15126, a bacteria that should not be prevalent in these samples. Upon further inspection, this genome had been tagged as 'misassembled' and repressed by NCBI. Therefore, we removed the genome from the kraken database and re-ran the program. The output of kraken was converted into a count matrix with taxa as rows and samples as columns, and the count for each cell representing the number of reads classified to that taxon, by sample. Taxa present in fewer than 10 samples were removed from further analysis to provide robust estimates of changes in abundance (N = 882 out of 3962 taxa). The count for each taxon was normalized within samples using CSS and a percentile of 0.5 (*Paulson et al., 2013*; *McMurdie and Holmes, 2014*), and normalized counts were aggregated to the species and genus levels.

## Analysis of log$_2$-fold change in abundance

In order to obtain a community-level view of the pattern of change from pre- to post-slaughter samples, multivariate, zero-inflated Gaussian mixture models were fit to species and genus-level normalized counts using metagenomeSeq's 'fitZig' function, with 'useCSSoffset' set to 'FALSE' as aggregation was performed with normalized counts (*Paulson et al., 2013*). All models included pen identification number as a covariable to account for potential clustering of observations. The output of fitZig was then transferred into limma's 'makeContrasts' and 'eBayes' functions to conduct pairwise comparisons of log$_2$-fold change in abundance between sample groups (*Smyth, 2004*), adjusting for multiple comparisons using the Benjamini-Hochberg procedure and using a critical $\alpha$ of 0.05.

## Acknowledgements

We thank JBS USA, LLC, JBS Five Rivers Cattle Feeding, LLC, and Drs. Tony Bryant and John Ruby (Greeley, CO) for providing access to abattoirs and feedlots; Santiago Luzardo, Megan Webb, Shuang Hu and Katie Rose McCullough for sampling assistance; and the University of Colorado Denver High Throughput Sequencing Core, which is supported in part by the Genomics and Microarray Shared Resource of Colorado's NIH/NCI Cancer Center Support Grant P30CA04693. This work was funded by the Beef Checkoff.

## Additional information

### Funding

| Funder | Grant reference number | Author |
| --- | --- | --- |
| National Institutes of Health | T32OD012201 | Noelle R Noyes |
| National Cattlemen's Beef Association | | Noelle R Noyes<br>Ifigenia Geornaras<br>Dale E Woerner<br>Paul S Morley<br>Keith E Belk |

The funders had no role in study design, data collection and interpretation, or the decision to submit the work for publication.

### Author contributions

NRN, Conception and design, Acquisition of data, Analysis and interpretation of data, Drafting or revising the article; XY, TAM, KLJ, Acquisition of data, Analysis and interpretation of data, Drafting or revising the article; LML, RJM, AD, SC, JR, Acquisition of data, Drafting or revising the article; IG, KEB, Conception and design, Acquisition of data, Drafting or revising the article; DEW, HY, Conception and design, Drafting or revising the article; SPG, CAB, Analysis and interpretation of data,

Drafting or revising the article; PSM, Conception and design, Analysis and interpretation of data, Drafting or revising the article

## Author ORCIDs

Noelle R Noyes, http://orcid.org/0000-0001-6149-1008

## Additional files

### Supplementary files

• Supplementary file 1. Sequencing, filtering and host removal statistics for all samples.

• Supplementary file 2. List of 319 ARDs identified across all 87 samples.

• Supplementary file 3. Resistance classification by class and mechanism

### Major datasets

The following datasets were generated:

| Author(s) | Year | Dataset title | Dataset URL | Database, license, and accessibility information |
|---|---|---|---|---|
| | 2015 | Beef Production Metagenome Raw sequence reads | http://www.ncbi.nlm.nih.gov/bioproject/292471 | Publicly available at the NCBI BioProject database (Accession no: PRJNA292471). |

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
