## [Decision Letter]

Thank you for submitting your work entitled "Resistome diversity in cattle and the environment decreases during beef production" for consideration by *eLife*. Your article has been favorably evaluated by a Senior editor, Richard Losick, and three reviewers, one of whom, Ben Cooper, is a member of our Board of Reviewing Editors.

The reviewers have discussed the reviews with one another and the Reviewing Editor has drafted this decision to help you prepare a revised submission.

All reviewers thought this was an interesting study and have made a number of constructive (and complementary) suggestions for improving the manuscript. In particular, all three reviewers highlighted aspects of the presentation that need to be substantially improved to make it clearer what was done (and why), and also to make it more accessible to a wider audience. This includes avoiding or defining jargon that won't be familiar to general readers. Two reviewers thought Figure 1 should be replaced by something more informative and this should be considered an essential revision.

Reviewer 1 highlighted the need to clarify which hypothesis tests were pre-planned (prior to examining the data) and which, if any, were post-hoc. Reviewer 1 also pointed out that where p-values are reported they should in general be accompanied by more informative summary measures. The section on fitness cost was also identified as needing attention.

Reviewer 2 highlighted the need to better present the relationship between AMU and the changes in ARD in the pen levels and also to explain the low number of ARG identified per sample, given that samples are pooled and expected to harbour millions of organisms. Reviewer 2 also made suggestions for improving Table 1 and requested a clear presentation of information about which pens received medicated feeds or injectable antibiotics.

Reviewer 3 highlighted the fact that, as presented, the work could potentially be misleading, with the implication that 1700 animals were followed rather than 8 cohorts. This reviewer also identified the need to discuss the limitations of metagenomics, in particular with reference to sensitivity limitations. A number of problems with Table 1 were also identified that need to be addressed.

All comments in the three reviews below should be addressed.

*Reviewer #1:*

As the first metagenomic investigation of AMR in intensive livestock systems this paper clearly addresses a subject of considerable importance and is likely to be of interest to a wide range of *eLife* readers.

I am not able to comment on many of the technical aspects relating to methods used in this paper; my comments come from the perspective of someone with related research interests (AMR in human populations).

Overall I found this work interesting but somewhat difficult to read. In part the difficulties relate to aspects of presentation which could be substantially improved, and I make a number of specific suggestions below. In part the difficulties arise from the use of methods with which I am not familiar (in particular, procrustes analysis); I strongly suspect this will be true of many other potential readers.

Generally, however, the methods do seem to be described well and with sufficient detail to allow repetition, even if the motivation for the analytical approaches taken is not always immediately clear. And as far as I could tell, analytical choices generally seem reasonable, and in most respects the conclusions seem to follow from the results. In a few places, however, I was concerned with an over-reliance on p-values. It wasn't clear if all hypotheses tests applied were pre-planned or whether some were suggested by the data (such as the test for a difference in the degree of correlation between ARD content and species composition in arrival and exit samples, as reported in the legend to Figure 4). If these are post-hoc (with comparison suggested by the data), the p-values won't be meaningful and it would be better to just report the magnitude of the difference (and, ideally, a measure of uncertainty) and flag up the observation as hypothesis generating. Equally, it wasn't clear to me if there were other hypotheses tests that were conducted but not reported. A clear statement of pre-planned hypothesis tests in the methods would be useful.

*Reviewer #2:*

This is an interesting study that addresses the dynamics of antimicrobial resistance determinants (ARD) during beef production in the USA. This area is still very underdeveloped and this paper could well make an important scientific contribution, both in terms of methodology as well as in learning about the biological processes behind. However, I found some formal deficiencies in the study, mostly in the Materials and methods and Results sections. Because of this, the paper as it stands reads a bit like a black-box study. I hope the authors can address some of the issues, detailed below, which could potentially improve the study greatly. Crucially, the authors should better present the relationship between AMU and the changes in ARD in the pen levels. The authors should provide some recommendation as to how to assess changes in resistome in other animal production systems in the USA or other countries. I find it a major flaw that sample results are very much dependent on the matrix typing, making comparison between different types of sample almost impossible. It is hard to understand the low number of ARG identified per sample (Cheng et al., 2012), given that samples are pooled and expected to harbour millions of organisms. The authors should provide plausible explanations for this. The prevalence of ARD per sample should be presented. Specific comments are provided below.

Introduction:

The background for the study should be re-worded with mention to the country and production systems so that the international audience understands the relevance and scope of the statements provided (i.e. consumption of beef, use of antimicrobials, etc.).

A reference should be provided for the statement immediately before Reference 2 (World Health Organization, 2011). The reference provided refers only to the WHO definition of antimicrobial of critical importance for human medicine.

Define 'feedlots'.

Use the metric system throughout (i.e. replace 1000 pounds by the decimal equivalent).

Materials and methods:

It is not clear to me whether the authors chose to investigate consistently ADD and/or 'number of days that a given AMD is expected to persist at therapeutic concentrations in the target tissue'. As far as I understand the first measure is the relevant one that reflects usage, and the second is an inherent feature of each individual antimicrobial compound.

Study design:

The study design is not properly described. Although the study does not seem to be a formal survey (only 4 feedlots investigated, 2 pens per feedlot), authors should provide data on the total population of feedlots in the USA, and strive to describe feedlot operations (i.e. number of cattle in each), so that the reader can get the idea of the scale of the study undertaken. The sampling plan is confusing. The should be conveniently be presented in a table indicating number of samples of each types from, as well as from how many sites (pens feces and floors, water, trucks, meat, trimming samples, end of conveyor belts, sponge samples). Some of these types of samples are described jargon language and therefore unclear for the non-expert (i.e. trimmings, 'holding pen floors', 'feedlot pen floors', 'truck surfaces', 'chuck' and 'round primal cuts'). The Figure 1 should be replaced by a flowchart that contains information about these sampling points, since the current figure is too basic and un-informative.

The sample processing description is very confusing. This should be explained following the same order as sample collection.

Results and Discussion:

Table 1: Since pens are the unit of sampling, the data should be presented both by pens and antimicrobial type (consider a two-way table), since the pens are the unit of the study.

First paragraph: The authors do not clearly present which pens had received medicated feeds or injectable antimicrobials. This information is important and should be presented.

Fifth paragraph: The statistical analyses for changes in prevalence of ARG corresponding to specific antimicrobials should be provided in Figure 5.

Sixth and seventh paragraphs: include anecdotal findings. The authors should provide their interpretation of these findings or otherwise delete.

*Reviewer #3:*

The authors provided a well-written and well-thought manuscript. My recommendation is to accept for publication with minor revisions.

I have 3 primary comments for the author to consider:

1) The authors imply that ~1700 animals were followed. However, given the pooling of the samples, they really only followed 8 cohorts and then an estimate of the cohort but sampling a small fraction of the feces, water, soil, etc. and meat derived from the animals. In that sense, little can be inferred to the individuals rather than some assessment of the cohort. As such, if a resistance gene is harbored by bacteria in some percentage of the animals, there is a probability of being including in the pool. No discussion of the value of a pooled measure of the cohort and potential variation among individuals is provided. All that is provided is a sense that the averages (i.e. pools of individuals) are similar.

2) Much discussion is afforded to the value of metagenomic assessment of the resistome in pooled samples. There is not much discussion of the limitation of metagenomics. Yet like all assays, there are limitations. Some aspects are implicitly discussed in terms of failing to detect ARD in post-slaughter samples. Granted abattoir interventions are reported to be very effective but it seems implausible that if one can perform a simple APC and recover bacteria post-slaughter that none harbor ARD. Hence there are sensitivity limitations that could be discussed in terms of sampling, DNA extraction/purification, amount of DNA amplified, etc.

3) Table 1 is flawed. In particular, tylosin is administered in-feed daily and the target site is the GIT yet the ADD is calculated for injectable tylosin with a target tissue of the lung. Simply developing an ADD based on a different formulation, route of administration, target tissue, target pathogen, etc. is flawed thinking. If the oral administration has an effect – and presumably it reduces the incidence of liver abscesses – then that should represent a defined daily dose. That is where an effect – if any – would be in terms of the GIT resistome. Why are some doses mg/lb and others lb/kg BW? Also, tylosin is applied to the pen so the probability of a treated animal being included in a pool is 100% yet only 14 animals were treated with a FQ. What is the probability of a FQ-treated animal being included in a pool? It if is low, then the resistome of a treated animal may look very different to the pen average? See points #1 and #2.

---

## [Author Response]

All reviewers thought this was an interesting study and have made a number of constructive (and complementary) suggestions for improving the manuscript. In particular, all three reviewers highlighted aspects of the presentation that need to be substantially improved to make it clearer what was done (and why), and also to make it more accessible to a wider audience. This includes avoiding or defining jargon that won't be familiar to general readers. Two reviewers thought Figure 1 should be replaced by something more informative and this should be considered an essential revision. Reviewer 1 highlighted the need to clarify which hypothesis tests were pre-planned (prior to examining the data) and which, if any, were post-hoc. Review 1 also pointed out that where p-values are reported they should in general be accompanied by more informative summary measures. The section on fitness cost was also identified as needing attention. Reviewer 2 highlighted the need to better present the relationship between AMU and the changes in ARD in the pen levels and also to explain the low number of ARG identified per sample, given that samples are pooled and expected to harbour millions of organisms. Reviewer 2 also made suggestions for improving Table 1 and requested a clear presentation of information about which pens received medicated feeds or injectable antibiotics. Reviewer 3 highlighted the fact that, as presented, the work could potentially be misleading, with the implication that 1700 animals were followed rather than 8 cohorts. This reviewer also identified the need to discuss the limitations of metagenomics, in particular with reference to sensitivity limitations. A number of problems with Table 1 were also identified that need to be addressed. All comments in the three reviews below should be addressed.

We appreciate that all three reviewers found this study to be of interest to *eLife* readers, and that all reviewers recognized the contribution of this work to the field of antimicrobial resistance research, particularly in the area of livestock production where there is relatively little published (as the reviewers point out).

We greatly appreciate the feedback received from all three reviewers, and feel that their comments have greatly improved the readability, clarity, and impact of the manuscript. We have addressed each reviewer’s comments in detail below. Here, we also list the major modifications made to the manuscript, at a summary level:

Figure 1 was modified to include study design information and sampling timeframes; and to visually provide definitions of terms that are less well known to readers who aren’t familiar with livestock production. We also included source data for Figure 1 that includes sampling dates for all pens, locations and matrices. We hope that this not only addresses the concerns with Figure 1, but also concerns about clarity in study design/sampling, as well as use of jargon terms.

We have minimized the use of jargon terms wherever possible; in instances where we cannot avoid their use (e.g., “feedlot”, “pen”, “trimmings”), we have explicitly defined them. We hope that this improves accessibility of the manuscript.

Throughout the manuscript, we have added sentences pertaining to the motivations behind study design, sampling and analysis decisions. We also added a section in the Methods that details which analyses were pre-planned versus post-hoc.

We modified Table 1 to provide pen-level AMD use information, as well as to obviate the need for use of ADDs (which were mentioned by 2 reviewers).

We included additional content and citations (in the Abstract, Results/Discussion and conclusion sections) to balance out the opportunities *and* limitations of the metagenomics approach.

We believe that these modifications have significantly clarified the study design, sampling and analysis, while also making the manuscript more accessible to the broad *eLife* readership. We thank you and all reviewers again for the very constructive feedback.

*Reviewer #1: I am not able to comment on many of the technical aspects relating to methods used in this paper; my comments come from the perspective of someone with related research interests (AMR in human populations). Overall I found this work interesting but somewhat difficult to read. In part the difficulties relate to aspects of presentation which could be substantially improved, and I make a number of specific suggestions below. In part the difficulties arise from the use of methods with which I am not familiar (in particular, procrustes analysis); I strongly suspect this will be true of many other potential readers.*

Thank you for pointing this out. We have added several sentences clarifying the motivation behind some of these less well-known methods (see responses to points #4 and #5, below).

*Generally, however, the methods do seem to be described well and with sufficient detail to allow repetition, even if the motivation for the analytical approaches taken is not always immediately clear.*

Thank you. We hope that we have addressed the motivation behind analytical approaches by including motivations behind specific analyses (e.g. Hellinger and procrustes), and also by indicating which analyses were planned versus post-hoc; please see below for a more detailed response to this concern. Please let us know if more clarification on motivation is needed.

And as far as I could tell, analytical choices generally seem reasonable, and in most respects the conclusions seem to follow from the results. In a few places, however, I was concerned with an over-reliance on p-values. It wasn't clear if all hypotheses tests applied were pre-planned or whether some were suggested by the data (such as the test for a difference in the degree of correlation between ARD content and species composition in arrival and exit samples, as reported in the legend to Figure 4). If these are post-hoc (with comparison suggested by the data), the p-values won't be meaningful and it would be better to just report the magnitude of the difference (and, ideally, a measure of uncertainty) and flag up the observation as hypothesis generating.

Thank you for making this important point; we agree that we do not want to over-emphasize p-values, especially since this is largely a descriptive study of the resistome in a relatively under-studied population/environment. We have attempted to correct for this overemphasis in several ways:

You are correct that the procrustes test to correlate ARD and species composition was post-hoc as a means to tease apart microbiome-driven resistome changes versus “independent” resistome changes. We have removed the p-value from the legend for Figure 4, and have included additional wording in the manuscript body to clarify that the comparison was post-hoc and to describe the motivation in more detail (Results and Discussion, third paragraph). We also changed some wording to indicate that the procrustes results led to a hypothesis that AMD exposures may have been driving resistome changes above and beyond the microbiome.

We have removed the p-value criterion (red shaded bubbles) from Figure 6 (now Figure 7); this figure was intended to show general microbiome trends from pre- to post-slaughter, and therefore the p-values were overkill. If readers want to see the model output that generated this figure, they can look into the source data file.

We have also removed references to p-values in the discussion of differences between NMDS ordination results between pens, feedlots and states; for interested readers, these are available as source data for the new Figure 6 (see below).

*Equally, it wasn't clear to me if there were other hypotheses tests which were conducted but not reported. A clear statement of pre-planned hypothesis tests in the methods would be useful.* Thank you for noting this omission; we now explicitly state throughout the Methods section which tests were pre-planned, and which were not. We also added a separate section to the Methods that clearly states which tests were pre-planned, and which were performed post-hoc (”Ordination and heatmap generation”).

*Reviewer #2: This is an interesting study that addresses the dynamics of antimicrobial resistance determinants (ARD) during beef production in the USA. This area is still very underdeveloped and this paper could well make an important scientific contribution, both in terms of methodology as well as in learning about the biological processes behind.However, I found some formal deficiencies in the study, mostly in the Materials and methods and Results sections. Because of this, the paper as it stands reads a bit like a black-box study. I hope the authors can address some of the issues, detailed below, which could potentially improve the study greatly.*

Thank you for prompting us to make the Methods and Results sections clearer. We have included additional details in both of these sections, as detailed below. We have also rewritten and reorganized several subsections to make them flow more logically. Finally, we have also attempted to minimize and define jargon terms, which should hopefully make the study more accessible. Please see below for more details regarding these modifications.

*Crucially, the authors should better present the relationship between AMU and the changes in ARD in the pen levels.*

Thank you for prompting us to make our results and conclusions on this association more clear. At a general level, this study was not designed to test hypotheses about specifics associations between AMU and ARDs at the pen level. We have added some sentences to make this clear (see lines Results and Discussion, first paragraph and Conclusion, last paragraph). While we do not feel comfortable drawing conclusions about associations between specific AMDs (or prevalence of use) and ARD levels, we have discussed high-level, ecological relationships in several places; in many of these sections, we have rewritten the content to make our conclusions more clear.

The section on decreased ARD diversity during the feeding period discusses the possibility that selection pressure from AMU could be driving the decreased diversity.

The section on the finding of carbapenemases, streptogramin and PhLOPS_A_ ARDs in feedlot samples explicitly states that these AMDs are not used in these study populations, and therefore “antimicrobial use practices cannot directly explain the presence of these important ARDs”.

We discuss the fact that the lack of difference between the resistome at the pen, feedlot and state levels may be due to the fact that AMU protocols are very similar within large North American feedlots.

*The authors should provide some recommendation as to how to assess changes in resistome in other animal production systems in the USA or other countries.*

Thank you for pointing out this omission. We have added relevant text in the Conclusion, first paragraph.

*I find a major flaw that sample results are so much dependent on the matrix typing, making comparison between different types of sample almost impossible.*

The large difference by matrix type is definitely something that we had to control for in analysis, and does make comparisons between samples challenging. We attempted to control for this by performing arrival-exit-holding comparisons on each matrix separately (Figure 2, Figure 3–Figure 3 and Figure 4). Figure 2 and Figure 5 did lump all of the sample matrices together, and we have changed these figures accordingly. Figure 2 now shows each matrix by histogram bin, and we have removed the Holding Pen samples from Figure 5 so that this figure only contains an “apples-to-apples” comparison (i.e., the arrival and exit samples each had the same number of samples from each matrix type). We have also modified the legend for Figure 5 so that the “N” for each matrix is broken out, and the reader can see explicitly that the number of samples of each type was the same. Unfortunately for Figure 6, it is impossible to break out all sample matrices because there is near-complete confounding between sample type and location (i.e., all of the post-slaughter samples are rinsates and trimmings, compared to feces, soil and water for most pre-slaughter samples). Therefore, we have kept this analysis in the paper, but we have also added an additional “apples-to-apples” comparison of sponge samples; this includes the sponge samples taken in the transport trucks (as the pre-slaughter set) and the sponge samples taken from the conveyor belts (as the post-slaughter set). These results are now reported in the text, in Figure 7—figure supplement 1 and its corresponding source data.

*It is hard to understand the low number of ARG identified per sample (Cheng et al., 2012), given that samples are pooled and expected to harbour millions of organisms. The authors should provide plausible explanations for this.*

Thank you for this comment. We have included several sentences contextualizing the number of ARD per sample in terms of related studies (both functional and shotgun metagenomic, human and cattle). When reading such literature, we find that we actually identified a relatively high number of ARD per sample, which could likely be due to pooling, as you point out. We have included these citations and text in the relevant section.

*The prevalence of ARD per sample should be presented.*

Thank you for requesting these data, which we agree should be presented. We have added this information to [Supplementary-material SD5-data], and have also included additional details in the beginning of the Results and Discussion section.

*Specific comments are provided below.*

*Introduction:*

*The background for the study should be re-worded with mention to the country and production systems so that the international audience understands the relevance and scope of thestatements provided (i.e. consumption of beef, use of antimicrobials, etc.).*

Thank you for pointing out these omissions. We have included relevant context in several places (Introduction, Results and Discussion). We also included a citation for CIPARS (Introduction, first paragraph, along with citation of NARMS).

*A reference should be provided for the statement immediately before Reference 2 (World Health Organization, 2011). The reference provided refers only to the WHO definition of antimicrobial of critical importance for human medicine.*

We have added FDA Guidance #152 to the reference list for this statement.

*Define 'feedlots'.*

We have included a short description of a feedlot in the first paragraph of the Introduction.

*Use the metric system throughout (i.e. replace 1000 pounds by the decimal equivalent).*

We have replaced pounds with kilograms in this specific instance, and have used SI units throughout, including in Table 1 for drug dosages.

Materials and methods:

*It is not clear to me whether the authors chose to investigate consistently ADD and/or 'number of days that a given AMD is expected to persist at therapeutic concentrations in the target tissue'. As far as I understand the first measure is the relevant one that reflects usage, and the second is an inherent feature of each individual antimicrobial compound.*

We have removed the ADD units from Table 1 and from the manuscript. ADDs are a unit meant to standardize dosages across animals and drugs, and therefore include both a measure of usage (the dose) AND the inherent pharmacokinetics of individual compounds within specific target tissues (and hosts). ADDs are primarily used to standardize treatment data so that associations between AMU and AMR can be compared across drugs, dosages and animals. However, since we are not performing statistical tests of AMU-AMR associations, we do not need to introduce ADDs into the analysis. Thank you for pointing this out.

Study design:

*The study design is not properly described. Although the study does not seem to be a formal survey (only 4 feedlots investigated, 2 pens per feedlot), authors should provide data on the total population of feedlots in the USA, and strive to describe feedlot operations (i.e. number of cattle in each), so that the reader can get the idea of the scale of the study undertaken.*

Thank you for pointing out this omission. We have added some statistics on the US feedlot population in the first paragraph of the Results and Discussion. We have also expanded our description of the selection strategy for the feedlots, pens, sampling locations and sample types.

The sampling plan is confusing. The should be conveniently be presented in a table indicating number of samples of each types from, as well as from how many sites (pens feces and floors-, water, trucks, meat, trimming samples, end of conveyor belts, sponge samples). Some of these types of samples are described jargon language and therefore unclear for the non-expert (i.e. trimmings, 'holding pen floors', 'feedlot pen floors', 'truck surfaces', 'chuck' and 'round primal cuts'). The Figure 1 should be replaced by a flowchart that contains information about these sampling points, since the current figure is too basic and un-informative.

Thank you for pointing out our use of jargon and the confusion around the sampling plan. Whenever we were unable to avoid using jargon, we have defined the word upon first use (i.e. pen, abattoir, holding pens, trimmings, chuck and round). We have also revised Figure 1 to hopefully be more informative in terms of the sampling plan, and also included in [Supplementary-material SD1-data] that includes a table of all of the sampling location, dates and types. Please let us know if this is not sufficient.

*The sample processing description is very confusing. This should be explained following the same order as sample collection.*

Thank you for pointing this out. We have made several modifications to make the sample processing section flow more logically: included subheaders for each step (pre-extraction, extraction and sequencing) and described each subsection in the same order as sample collection (feces, soil, water, sponge, rinsates).

Results and Discussion:

Table 1: Since pens are the unit of sampling, the data should be presented both by pens and antimicrobial type (consider a two-way table), since the pens are the unit of the study.

Thank you for pointing this out; we have changed Table 1 accordingly.

*First paragraph: The authors do not clearly present which pens had received medicated feeds or injectable antimicrobials. This information is important and should be presented.*

We agree, and we have changed Table 1 so that it now indicates how many animals received which drugs, by pen.

Fifth paragraph: The statistical analyses for changes in prevalence of ARG corresponding to specific antimicrobials should be provided in Figure 5.

Figure 5 does not include any statistical analyses, but rather depicts a descriptive analysis of the proportion of samples at each sampling location that were positive for at least 1 ARD within each listed mechanism (which are grouped by class). Figure 5 has been modified to exclude holding pen results, since – as you point out – there was confounding by matrix.

*Sixth and seventh paragraphs: include anecdotal findings. The authors should provide their interpretation of these findings or otherwise delete.*

Thank you for your comment. We have provided our interpretation of these findings in the Results and Discussion section.

*Reviewer #3: The authors provided a well-written and well-thought manuscript. My recommendation is to accept for publication with minor revisions.*

*I have 3 primary comments for the author to consider:*

*1) The authors imply that ~1700 animals were followed.*

Thank you for noting this; we agree that, as written, the Abstract implies that ~1700 individual animals were tracked. We have changed the Abstract accordingly.

*However, given the pooling of the samples, they really only followed 8 cohorts and then an estimate of the cohort but sampling a small fraction of the feces, water, soil, etc. and meat derived from the animals. In that sense, little can be inferred to the individuals rather than some assessment of the cohort. As such, if a resistance gene is harbored by bacteria in some percentage of the animals, there is a probability of being including in the pool. No discussion of the value of a pooled measure of the cohort and potential variation among individuals is provided. All that is provided is a sense that the averages (i.e. pools of individuals) are similar.*

Thank you for highlighting this important point. We have included relevant content in the subsection “Sample size estimation and sampling design”; Results and Discussion, first paragraph; and Conclusion, last paragraph.

*2) Much discussion is afforded to the value of metagenomic assessment of the resistome in pooled samples. There is not much discussion of the limitation of metagenomics. Yet like all assays, there are limitations.*

Thank you for emphasizing this important point; we do not wish to give a biased report of this approach. We have added the word “limitations” to the Abstract, and also included additional content in the conclusion regarding the limitations of the metagenomics assay in regards to ARD identification (Conclusion, second paragraph). In addition, we have expanded the discussion of the LOD of metagenomics (Results and Discussion, last paragraph and Conclusion, first paragraph). We have also addressed the inability of metagenomics to investigate expression of ARDs in the paragraph describing ARDs against critically important antimicrobials (Results and Discussion, sixth paragraph).

*Some aspects are implicitly discussed in terms of failing to detect ARD in post-slaughter samples. Granted abattoir interventions are reported to be very effective but it seems implausible that if one can perform a simple APC and recover bacteria post-slaughter that none harbor ARD. Hence there are sensitivity limitations that could be discussed in terms of sampling, DNA extraction/purification, amount of DNA amplified, etc.*

Thank you for making this important point. We have added additional wording in the Results and Discussion, eighth paragraph, that specifically addresses the APC issue, and also discusses the Se limitations of the metagenomics assay. This is a tricky issue, as the available data suggests that APC counts coming off of post-chilled carcasses are commonly below LOD (i.e. 10^0^ or 10^1^ CFU/100cm^2^). However, we then tend to see an uptick in APCs at the end of the fabrication line, suggesting that bacteria are somehow either re-introduced into carcasses, or the very, very rare existing bacteria undergo rapid growth during the fabrication process. The increased surface area that occurs during fabrication likely contributes as well. In any case, we agree with the need to address the apparent discrepancy between APC counts and our results, and have added citations of APCs for post-fabrication products; we would point, however, that the only publicly available citations are relatively outdated (2007), and intervention strategies have changed in the meantime. In short, more work is needed to understand the LOD for metagenomics in high-host background samples, as we point out in the Results and Discussion section, last paragraph and the Conclusion, first paragraph.

*3) Table 1 is flawed. In particular, tylosin is administered in-feed daily and the target site is the GIT yet the ADD is calculated for injectable tylosin with a target tissue of the lung. Simply developing a ADD based on a different formulation, route of administration, target tissue, target pathogen, etc. is flawed thinking. If the oral administration has an effect – and presumably it reduces the incidence of liver abscesses* – *then that should represent a defined daily dose. That is where an effect* – *if any* – *would be in terms of the GIT resistome.*

Thank you for this very considered comment on the ADD value for in-feed tylosin. Based on your comment, as well as Reviewer #2, we have removed ADD calculations completely from this table. Because we were not undertaking a formal analysis of associations between AMU and AMR, we felt that we did not need to present ADDs, but rather just number of doses; this also removes any differences or concerns about appropriate ADD values for different drug dosages and uses. We have also broken out treatment data by pen in order to allow readers to better understand AMU distributions across pens.

*Why are some doses mg/lb and others lb/kg BW?*

We were using the labeled dosage for the brand name for this table; we have converted mg/lb to mg/kg BW.

Also, tylosin is applied to the pen so the probability of a treated animal being included in a pool is 100% yet only 14 animals were treated with a FQ. What is the probability of a FQ-treated animal being included in a pool? It if is low, then the resistome of a treated animal may look very different to the pen average? See points #1 and #2.

Thank you for pointing out this possible scenario. We have now addressed this concern in multiple areas, including the Conclusion and the Materials and methods (sample size estimation and sampling design subsection).